# A three-dimensional phase-field model for multiscale modeling of thrombus biomechanics in blood vessels

**Xiaoning Zheng**[1], **Alireza Yazdani**[1], **He Li**[1], **Jay D. Humphrey**[2], **George E. Karniadakis**[1]*

**1** Division of Applied Mathematics, Brown University, Providence, Rhode Island, United States of America,
**2** Department of Biomedical Engineering, Yale University, New Haven, Connecticut, United States of America

* george_karniadakis@brown.edu

**Data Availability Statement:** All relevant data are within the manuscript and its Supporting Information files.

## Abstract

Mechanical interactions between flowing and coagulated blood (thrombus) are crucial in dictating the deformation and remodeling of a thrombus after its formation in hemostasis. We propose a fully-Eulerian, three-dimensional, phase-field model of thrombus that is calibrated with existing *in vitro* experimental data. This phase-field model considers spatial variations in permeability and material properties within a single unified mathematical framework derived from an energy perspective, thereby allowing us to study effects of thrombus microstructure and properties on its deformation and possible release of emboli under different hemodynamic conditions. Moreover, we combine this proposed thrombus model with a particle-based model which simulates the initiation of the thrombus. The volume fraction of a thrombus obtained from the particle simulation is mapped to an input variable in the proposed phase-field thrombus model. The present work is thus the first computational study to integrate the initiation of a thrombus through platelet aggregation with its subsequent viscoelastic responses to various shear flows. This framework can be informed by clinical data and potentially be used to predict the risk of diverse thromboembolic events under physiological and pathological conditions.

## Author summary

Thromboembolism is associated with detachment of small thrombus pieces from the bulk in the blood vessel. These detached pieces, also known as emboli, travel through the blood flow and may block other vessels downstream, e.g., they may plug the deep veins of the leg, groin or arm, leading to venous thromboembolism (VTE). VTE is a significant cause of morbidity and mortality and it affects more than 900,000 people in the United States and result in approximately 100,000 deaths every year. Mechanical interaction between flowing blood and a thrombus is crucial in determining the deformation of the thrombus and the possibility of releasing emboli. In this study, we develop a phase-field model that is capable of describing the structural properties of a thrombus and its biomechanical properties under different blood flow conditions. Moreover, we combine this thrombus

**Funding:** This work was supported by grant U01 HL 1163232 and U01 HL 142518 of National Institute of Health [https://www.nih.gov] (J.H. and G.K.). The funders had no role in study design, data collection and analysis, decision to publish, or preparation of the manuscript.

**Competing interests:** No authors have competing interests.

model with a particle-based model which simulates the initiation of the thrombus. This combined framework is the first computational study to simulate the development of a thrombus from platelet aggregation to its subsequent viscoelastic responses to various shear flows. Informed by clinical data, this framework can be used to predict the risk of diverse thromboembolic events under physiological and pathological conditions.

## Introduction

Under physiological conditions, blood clots form at sites of vascular injury to prevent blood loss, but they eventually resolve as the vascular wall heals [1, 2]. The situation can be very different in pathological cases, however, in aortic dissection, which represents a severe injury to the vessel wall that manifests as a delamination that propagates through the media [3], a key clinical question is whether the thrombosis in the false lumen will remain biologically active or be resolved or remodeled as part of the healing process. Other pathological conditions, such as deep vein thrombosis [4–6], pulmonary embolism [7, 8], and atherothrombosis [9, 10], exhibit excessive and undesirable thrombi within the lumen that can result in partial or complete thrombotic vessel occlusion. Thrombus formation is a multiscale process [11–13]: upon a vascular injury or adverse hemodynamics, freely flowing platelets in the blood become activated and form aggregates that cover the thrombogenic area within tens of seconds. Subsequently, a series of biochemical reactions [14, 15], can contribute to the formation of fibrin from the blood-borne fibrinogen. Fibrin monomers form a network that strengthens the platelet aggregates and facilitates thrombus maturation within minutes to hours.

Hemodynamics is essential in the process of thrombus growth [16–18]: for example, prolonged residence time of blood cells in regions of weak flow promotes the accumulation of these cells, while high shear rates on the surface of a growing thrombus may alter the morphology of the thrombus and inhibit its expansion into the flow field. If not physiologically degraded, thrombus can begin to undergo a progressive remodeling with replacement of fibrin with collagen fibers [19–22], causing a dramatic increase in stiffness, strength, and stability [22, 23]. The mechanical integrity of a thrombus plays an important role in many pathologies. In deep vein thrombosis [4–6], growth of highly stable thrombus can cause complete vessel occlusion, whereas formation of an unstable thrombus increases the possibility of thromboembolism—a process where pieces of thrombus detach from their original site, transit with blood, and block distal blood vessels—causing life-threatening complications [24–26]. Therefore, a comprehensive understanding of the structural constituents of a thrombus and the corresponding mechanical properties is vital to predict thrombus shape and deformation under various hemodynamic conditions and to evaluate the risk of thromboembolism and other pathological consequences.

Extensive experimental studies have increased our understanding of the growth, deformation, and embolization or resolution of a thrombus under physiological and pathological flows [27–30]. These studies demonstrated that the deformation of a thrombus and its propensity to generate emboli depend strongly on thrombus mechanical properties and the hemodynamics. Whereas, there is a lack of *quantitative* information for thrombus deformation and embolization from *in vivo* studies [31], computational modeling can complement experimental studies by simulating thrombus formation while including sub-processes such as platelet aggregation and coagulation kinetics [32–35], as well as thrombus biomechanics in flow [31, 36–38]; see also reviews [39–42]. There are, however, few studies dedicated to quantifying the integrated process of thrombus initiation and development as well as its deformation and possible

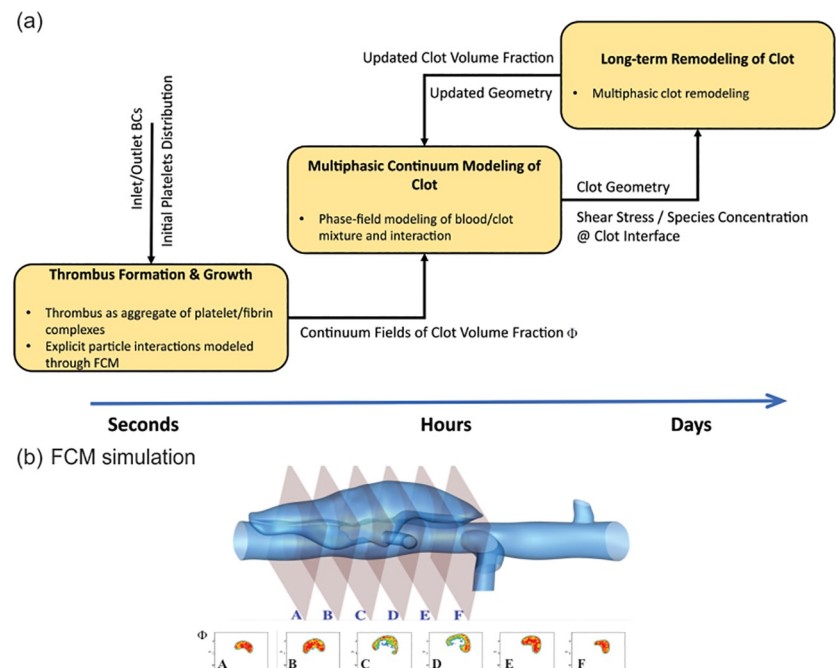

**Fig 1. Illustration of the simulation methodology.** (a) A multiscale computational framework to simulate the initiation, growth, and remodeling of a thrombus. (b) Volume fraction $\Phi_{fcm}$ field for 6 planar cross-sections in the false lumen of a dissecting aortic aneurysm taken along the flow direction from A to F. The platelet Lagrangian distribution in the aggregate obtained from FCM simulation is converted into continuum fields of thrombus volume fraction, which serves as an input for the phase-field method to simulate the interaction between flowing blood and thrombus. Figs. (a) and (b) are adopted from [3].

embolization under different hemodynamic conditions due to two primary computational challenges: first, thrombus formation is a slow biological process, with the time scale of platelet aggregation on the order of seconds while that of clot remodeling is on the order of days to weeks; second, a constitutive equation describing poro-viscoelastic properties of blood clots with different fibrin concentrations is lacking. Experimental investigations [29, 43, 44] show that a mature thrombus consists of a central core filled with densely packed fibrin and activated platelets. This core is cove with a shell consisting of loosely packed and partially activated platelets, allowing interstitial blood flow through. Hence, a thrombus can be considered as porous medium exhibiting viscoelastic behavior with a fibrous network contributing to both the viscous and the elastic properties [31, 45, 46].

In contrast to previous computational studies, which primarily focused on modeling either platelet aggregation or deformation of an existing thrombus, we present a framework that can capture the integrative process of thrombus formation and its subsequent deformation with possible generation of emboli under dynamic shear flow conditions. We simulate the aggregation of platelets by coupling a spectral/hp element method (SEM) [47] with a force coupling method (FCM) [48], following the strategy in [3, 33]. Once the platelets aggregate becomes stable, we convert the coarse-grained platelet distribution into a three-dimensional (3D) continuum field to estimate the clot volume fraction and continue the simulation of thrombus deformation within a blood flow using a phase-field model (see Fig 1), assuming that further formation of thrombi during the phase-field simulation. The parameters in the phase-field thrombus model are calibrated using existing *in vitro* experimental data. The connection between FCM particle and phase-field continuum simulations is facilitated by the volume

fraction of the thrombus, which is extracted from the FCM simulation as a continuum field; it serves as the initial condition for the subsequent phase-field modeling. As a demonstration of the capability of the proposed framework (FCM + phase-field), we simulate thrombus formation and subsequent interactions with flowing blood in a dilated circular vessel, mimicking a blood vessel with a fusiform aortic aneurysm.

This paper is organized as follows: in Results section, we study thrombus biomechanics while varying its permeability and blood flow-induced shear stresses. Subsequently, we calibrate the model parameters that control the permeability and calibrate viscoelastic properties of thrombus using *in vitro* experimental data. After that, we employ the phase-field model to simulate thrombus deformation in an idealized aneurysm following the initial formation and growth of the platelet aggregate captured by the FCM model. In Discussion section, we discuss the results, limitations, and strengths of the model. In Methods section, we detail the methodology for the proposed 3D framework including the FCM for simulating platelet aggregation and the phase-field method for simulating subsequent interactions between flowing blood and a thrombus.

## Results

### Effects of thrombus permeability and hemodynamics on thrombus deformation

The relative densities of cross-linked fibrin (fibrous) and platelet aggregates (granular) dictate thrombus permeability. Prior studies show that thrombus permeability and local blood flow-induced shear stresses play important roles in both the deformation of [31] and interstitial flows within thrombus [49]. To examine particular effects of permeability and hemodynamics on thrombus deformation, we first perform 2D and 3D simulations of blood flow passing a semi-spherical porous obstacle that consists of a shell and a core region, (see Fig 2), following the setup used by [31].

### Thrombus deformation in 2D rectangular and 3D cuboid channels

First, we compare simulation results between a 2D thrombus model based on a vector potential $\psi$ (see Eq 1 in Supporting information) and the 3D model based on the deformation

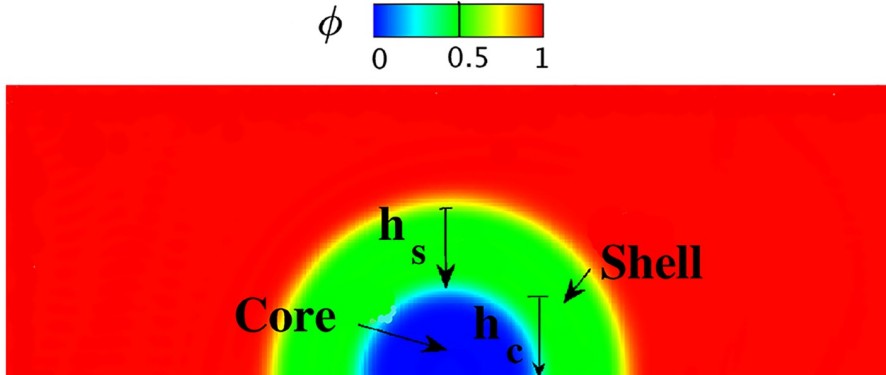

**Fig 2. Setup of the simulation for modeling thrombus deformation in a 2D rectangular channel (domain $\Omega = \{(x, y)| \, 0 \leq x \leq 6, 0 \leq y \leq 2\}$).** The initial volume fraction distribution of the blood/thrombus is denoted by the phase-field profile, with the core less permeable than the shell. $h_s$ and $h_c$ are dimensions of the shell and core regions of the thrombus, respectively.

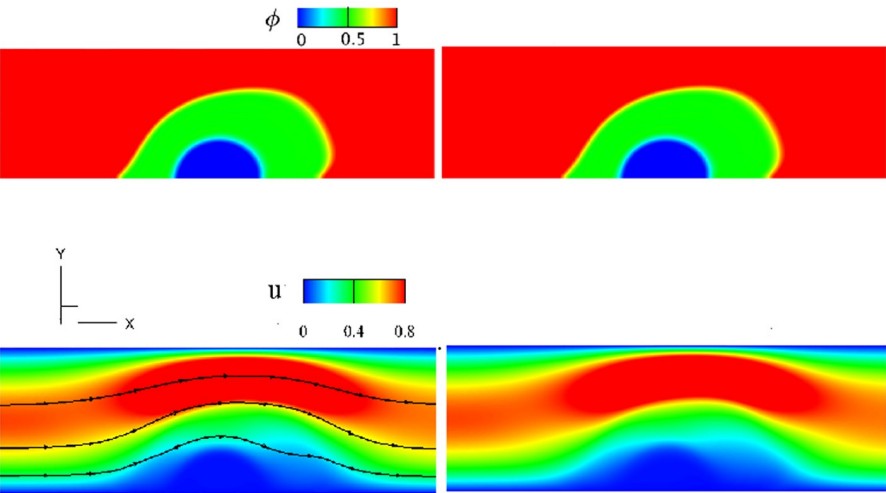

**Fig 3. Comparison of simulation results between 2D and 3D models.** The 3D channel has periodic boundaries in the $z$-direction. The phase-field and streamwise u-velocity are viewed in an $xy$-plane at time $T = 0.96$, specifically at $z = 0.5$ in the 3D simulation. First row: (a) 2D (b) 3D phase-field $\phi$, with $\phi = 1$ for the flowing blood and $\phi = 0$ for the thrombus; second row: (c) 2D (d) 3D $u$-velocity fields around the deformed thrombus. The shell permeability $\kappa_s = 10000$, the core permeability $\kappa_c = $ 1e-4, and the elastic shear modulus for the thrombus $\lambda_e = 0.5$.

gradient tensor **F** (see Eq 5) implemented in the present work. The initial setup of the 3D simulation is similar to that for the 2D simulation shown in Fig 2 except that we extend the 2D geometry into the $z$-direction with a thickness of 1 and impose a periodic boundary condition in the $z$-direction. The geometric and mechanical parameters for both simulations are: the thickness of the shell $h_s = 0.8$, the thickness of the core $h_c = 0.6$, the elastic shear modulus of the thrombus $\lambda_e = 0.5$, the density ratio of thrombus to blood $\rho_2/\rho_1 = 2$, the viscosity ratio of thrombus to blood $\eta_2/\eta_1 = 1$, and the maximum inlet velocity 0.75. As shown in Fig 3, under a steady flow, for both the 2D and 3D simulations, the shell region of the thrombus deforms more toward the downstream direction of the blood flow than the core region since the shell is more permeable to the blood flow than the core. Also, Fig 3(a)–3(d) show that the phase-field and velocity contours obtained from the 3D simulation at cross-section z = 0.5 are similar to those from the 2D simulation, although they are computed using different mathematical formulations. We also plot the time history of the streamwise $u$-velocity at three different locations as well as the $L^2$ norm of the $u$-velocity field for 2D and 3D simulations in Fig 4, which suggests that the simulation results for the 2D and 3D models are consistent. These results confirm that the general 3D formulation can reduce to the 2D formulation when the third direction is homogeneous with periodic boundary conditions.

## Effect of thrombus permeability on the flow field in a circular vessel

Here we examine thrombi with different permeabilities. The computational domain is a circular vessel with a unit diameter and length of 10. A semi-spherical thrombus composed of a core and shell is initially placed in the lumen (see Fig 5). The flow profile at the inlet of the vessel is set to be parabolic with a maximum value of 0.75 and a zero-Neumann boundary condition is prescribed at the outlet. All relevant model parameters are given in the first row of Table 1. To isolate the effect of the permeability $\kappa_s$, we do not consider elastic energy in the permeability tests (i.e., the thrombus is rigid). The core permeability $\kappa_c = 1e − 4$ is fixed for these simulations. Fig 5 shows the axial velocity field for different values of shell permeability

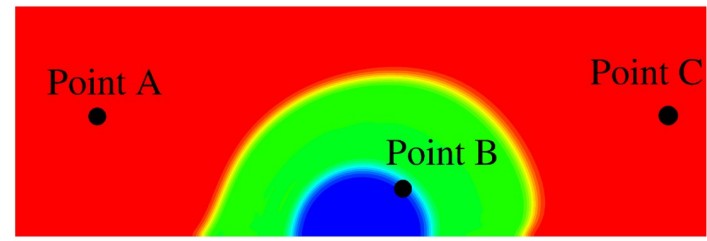

(a) Three points A,B,C for 2D and 3D velocity comparison.

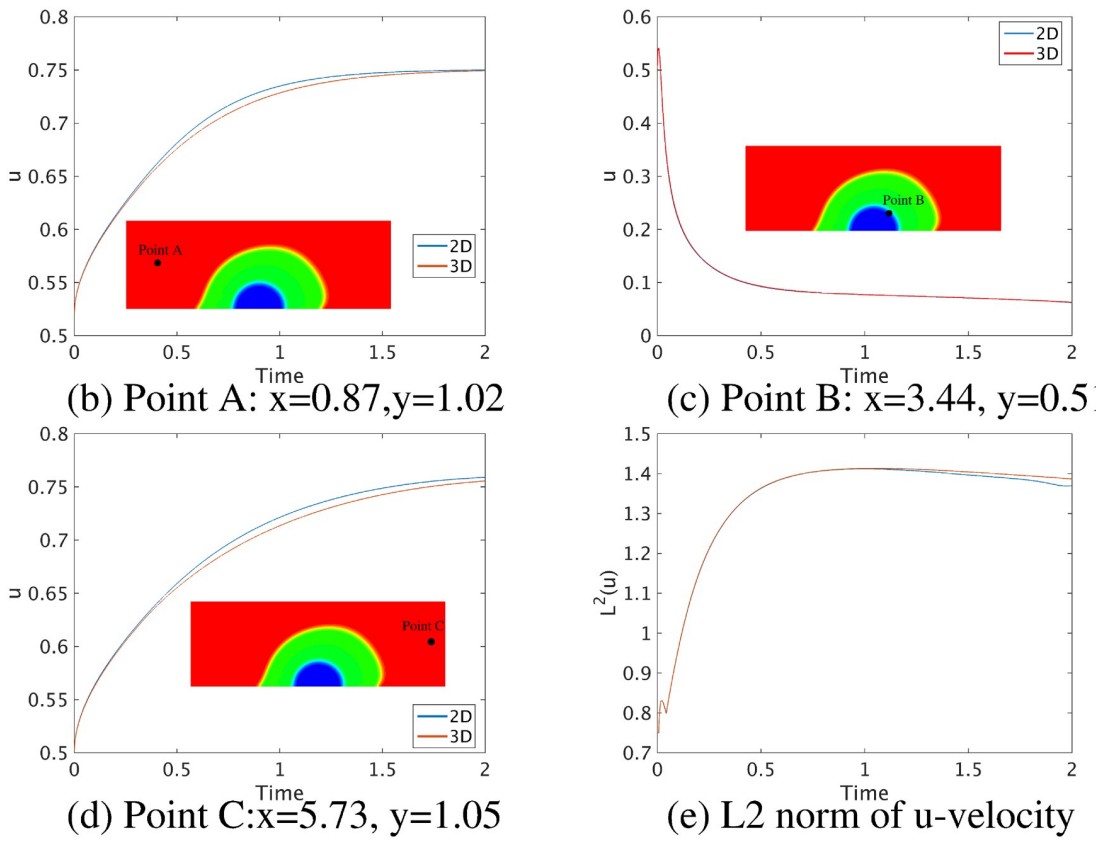

**Fig 4. Comparison of velocity magnitudes computed from 2D and 3D rectangular models; see Figs 2 and 3.** (a) Three points A, B, C for 2D and 3D velocity comparison. (b)-(d) Time-history of velocity $u$ at three different axial locations in the channel, and (e) $L^2$ norm of velocity $u$ computed from 2D and 3D models.

$\kappa_s$ of thrombus in $xy$ and $xz$ planes; it is observed that, blood velocity outside the thrombus increases significantly as shell permeability decreases, suggesting that the microstructure of the thrombus plays an important role on its neighboring flow field.

## Effect of hemodynamics on the deformation of a thrombus in a circular vessel

Next, we investigate the effect of hemodynamics on the deformation of a thrombus in a circular vessel. The same simulation setup as the permeability tests is used except that the elastic energy of the thrombus is included in the total energy relation. The parameters used in these

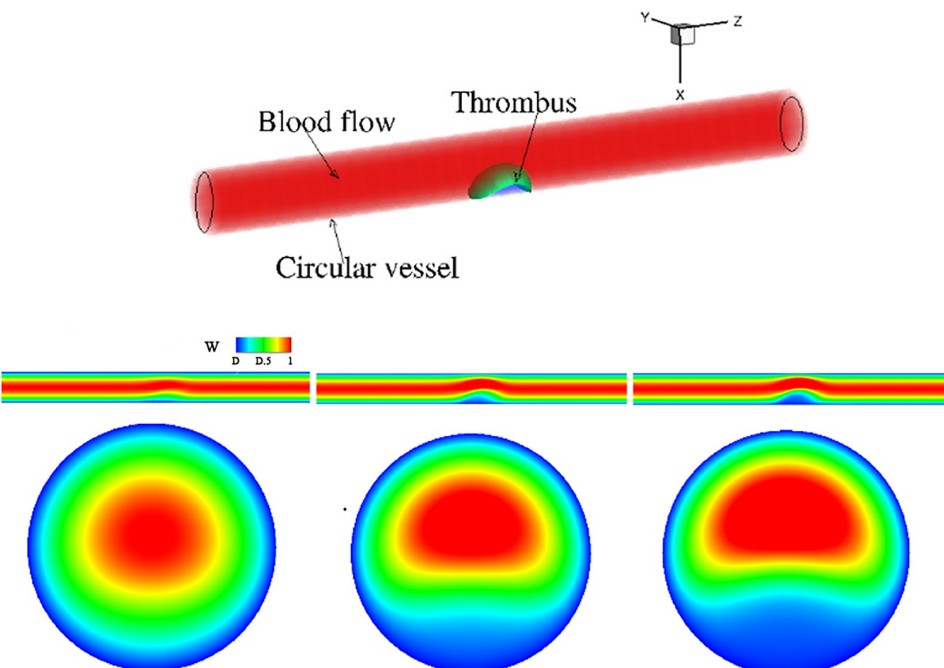

**Fig 5. Velocity field in a circular vessel with a thrombus inside.** (a) 3D view of simulation setup. Second row: Axial velocity field $w$ shown in the $xz$–plane cut at $y = 0$. Third row: a cross-sectional $xy$–plane cut through the center of thrombus. Here, $h_s = 0.5$, $h_c = 0.3$ and $\kappa_c = 1e - 4$ for all cases. The permeability of the thrombus varied as: (b) $\kappa_s = 1$, (c) $\kappa_s = 0.01$, (d) $\kappa_s = 0.005$ at time $T = 0.9$.

simulations are given in the second row of Table 1. Fig 6 shows that as the blood velocity increases, the deformation of the shell region becomes more dramatic whereas the deformation of the core region is not significant due to the increased flow velocity. These observations are consistent with prior experimental observations [29, 50] and numerical studies [31].

## Calibration of thrombus permeability and viscoelastic properties

In the previous section, we did not consider the connection between thrombus local volume fraction $\phi$ and its permeability $\kappa$ or the connection between $\phi$ and its elastic properties $\lambda_e$ when simulating the deformation of an idealized thrombus. In this section, we define these connections and calibrate the thrombus permeability and elastic properties by comparing our simulation results with existing experimental data.

### Calibration of the permeability of thrombus model using fibrin gel data

In this section, we use the fibrin gel data reported in [49] to calibrate the permeability of the thrombus model $\kappa(\phi)$ because of the limited experimental data on thrombi. As shown in Fig 7(a) fluid from a reservoir flows through a fibrin gel in a permeation chamber. The pressure is

**Table 1. Parameters used for thrombus deformation simulations in a circular vessel in non-dimensional units.**

| Parameters | $\frac{\rho_2}{\rho_1}$ | $\frac{\eta_2}{\eta_1}$ | $h_s$ | $h_c$ | $\lambda_e$ | $\sigma$ | $\kappa_s$ | $\kappa_c$ |
|---|---|---|---|---|---|---|---|---|
| | 10 | 2 | 0.5 | 0.3 | 0 | 1e-3 | 0.005,0.01,1 | 1e-4 |
| | 10 | 2 | 0.5 | 0.3 | 0.5 | 1e-3 | 1e-2 | 1e-4 |

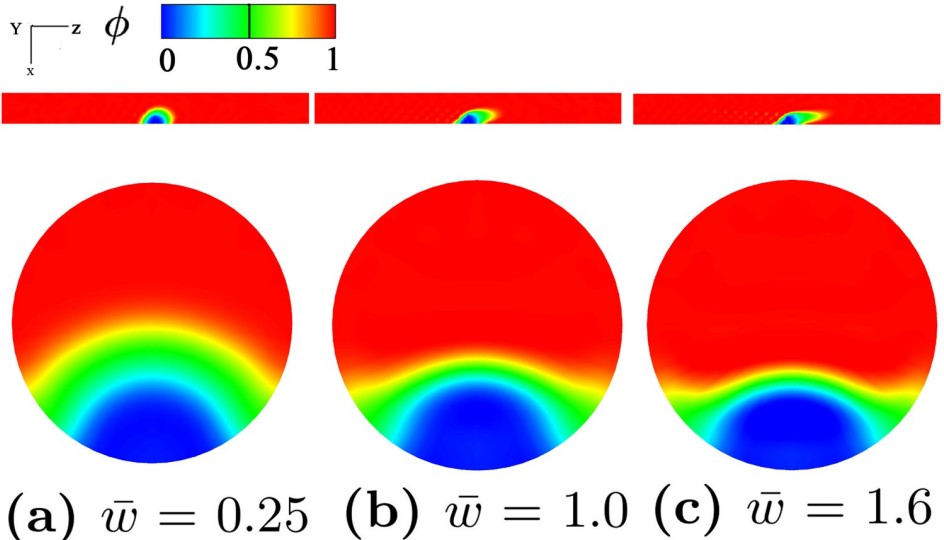

**Fig 6. Thrombus shapes in a vessel under different flow conditions.** The geometric parameters for thrombus shape are $h_s = 0.5$ and $h_c = 0.3$. Phase-field profile at different blood velocity: (a) $\bar{w} = 0.25$, (b) $\bar{w} = 1.0$, (c) $\bar{w} = 1.6$ shown at time $T = 0.91$. Phase-field plots shown on $xz$–plane at $y = 0$ in the first row and on $xy$–plane in the second row. $\bar{w}$: Average velocity in $z$-direction. $\phi = 1$ denotes flowing blood and $\phi = 0$ denotes solid thrombus.

measured before and after the chamber, and the flow rate is calculated by measuring the displacement of the air-buffer interface in the tubing at regular time intervals. The permeability $\kappa$ is then calculated using Darcy's law $v = -\kappa\Delta P/\eta$, where $v$ is the interstitial fluid velocity, $\eta$ is the viscosity of the percolating fluid, and $\Delta P$ is the pressure drop.

To mimic this experiment, we place a thrombus in the center of a straight cuboid channel filling the channel, as shown in Fig 7(b). The computational domain is $\Omega = \{(x, y, z)| 0 \leq x \leq 6, 0 \leq y \leq 2, 0 \leq z \leq 1\}$, and no-slip boundary conditions are applied to the upper and lower walls. Blood flow with a parabolic velocity profile is imposed at the inlet of the channel and, a zero-Neumann boundary condition is imposed at the outlet. We use the correlation $\kappa(\phi)/a_f^2 = [16\Psi_f^{1.5}(1 + 56\Psi_f^3)]^{-1}$ [49], also known as Davies' equation, to calculate the permeability. We vary the initial thrombus volume fraction, and simulate nine cases to calculate the pressure drop of the flow passing through thrombus. Four initial simulation setups are shown in Fig 7(c)–7(f). Parameters are density ratio $\rho_2/\rho_1 = 1$, viscosity ratio $\eta_2/\eta_1 = 2$, length $L = 0.76mm$, and width $H = 0.25mm$. The maximum inlet velocity $v_{max}$ is $1mm/s$, which gives a Reynolds number $Re = \rho_1 v_{max}H/\eta_1 = 0.02$. Furthermore, the average volume fraction (VF) is calculated by $\int_\Omega [(1 - \phi)/2]dv/V_{total}$. The velocity field and the phase-field contour for the case of VF = 0.66 are plotted in Fig 7(g) and 7(h), respectively, which shows a notable deformation of the thrombus. Our simulation results in Fig 7(i) show that the permeability decreases significantly with the increased VF. We also observe a good agreement between our simulation results and both the experimental data and the empirical fit (Davies' equation) [49].

## Calibration of thrombus elastic shear modulus $\lambda_e$

In our model, the elastic shear modulus $\lambda_e$ depends on the local volume fraction of a thrombus, which is equivalent to the phase-field variable $\phi$ in this study. There are few rheometry experiments that used oscillatory shear deformations to measure the viscoelastic properties of thrombus as a function of fibrin concentration, although they did not cover a wide range of fibrin

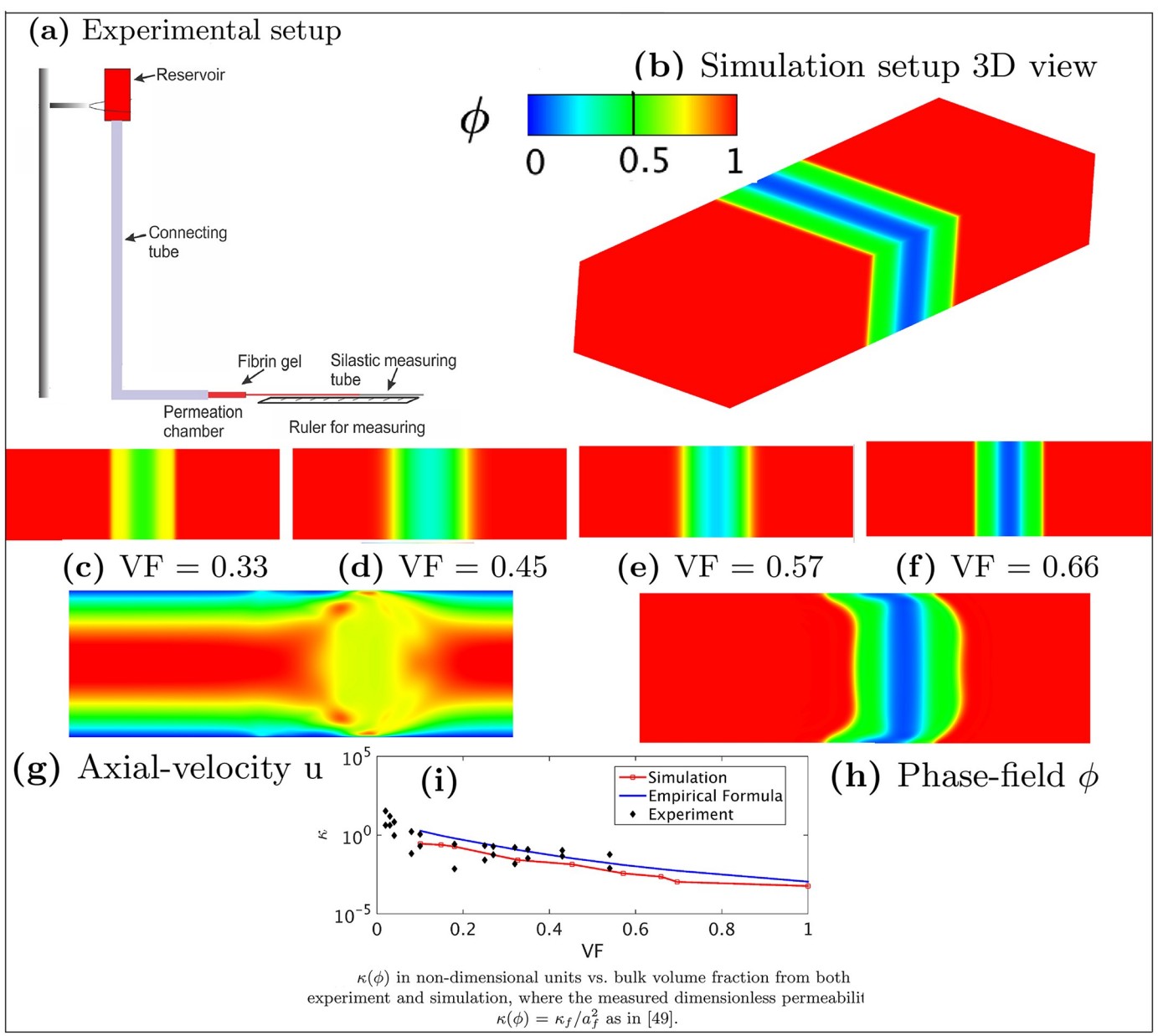

**Fig 7. Calibration of thrombus permeability κ.** (a) Experimental setup for calibrating thrombus permeability, a typical experimental setup contains a reservoir connected to the permeation chamber via a connecting tube. (b) 3D view of the simulation setup. (c-f) Four initial structures of the thrombus models with VF = 0.33, 0.45, 0.57, 0.66, respectively. (g) Axial-velocity $u$ in the vessel with thrombus VF = 0.66. (h) Phase-field $\phi$ in the vessel with thrombus VF = 0.66. (i) $\kappa(\phi)$ as a function of the bulk thrombus VF in logarithmic scale. Red line represents the simulation results of the phase-field model. Black dots represent the experimental data as reported in [49]. Blue line represents the relation expressed by Davie's Equation in [49].

concentration. In [51], a small oscillatory shear deformation with amplitude of $\gamma_0 = 0.01$ and frequency $f = 0.5\ Hz$ was used to quantify the viscoelasticity of a fibrin gel at low concentrations, which translates to a small thrombus volume fraction. The oscillatory deformation is sufficiently small to probe the linear viscoelastic regime using a Kelvin-Voigt model that decomposes shear stress into elastic and viscous contributions. Furthermore, the storage $G'$ and loss $G''$ moduli were quantified for the fibrin gel by the experiment. Both moduli may be combined into a phase angle $\delta = \tan^{-1}(G''/G')$ that describes the phase shift between the

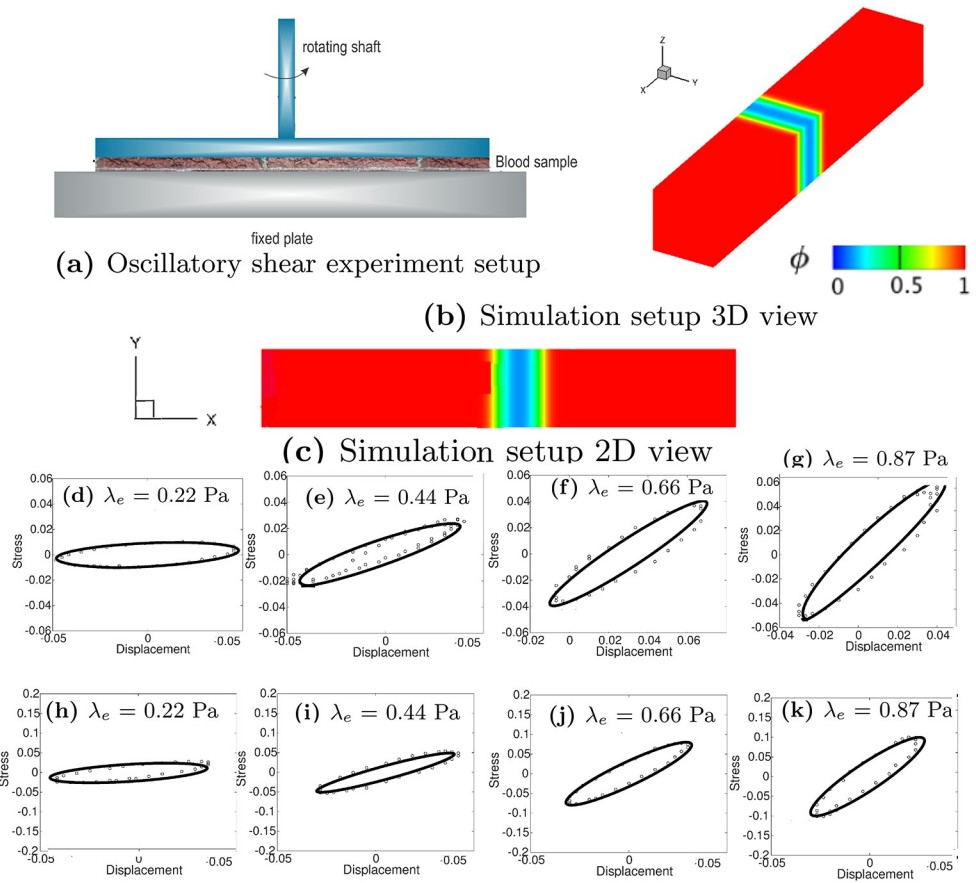

**Fig 8. Calibration of thrombus elastic shear modulus $\lambda_e$.** (a) Experimental setup for a typical oscillatory shear test. (b) 3D view of the simulation setup. (c) 2D view of the simulation setup. Third and Fourth rows: Stress-displacement loop for different $\lambda_e$. Quantities are all in non-dimensional units. Third row (d-g): VF (volume fraction) = 0.3419; Fourth row (h-k): VF = 0.5129. Points are data and lines are the fits. Displacement and stress are nondimensionalized with the characteristic velocity and length scales $3.33 \times 10^{-3}$ $m/s$ and $3 \times 10^{-4}$ $m$, respectively.

imposed strain and the measured stress. The two moduli are related by

$$G^{''} = 2\pi f(\eta_1 + \lambda_s G'), \tag{1}$$

where $f$ is the frequency of the oscillations, $\eta_1$ is the fluid viscosity, and $\lambda_s$ is the relaxation time defined as $\lambda_s = \eta_2/\lambda_e$ with $\eta_2$ the viscosity of the fibrin gel. Eq 2 describes how the storage and loss moduli are calculated, i.e.,

$$\delta = \sin^{-1}(\frac{4A_r}{\pi \Delta T_s \Delta d}), \quad G' = \frac{\Delta T_s}{\Delta d} \, \cos(\delta), \quad G^{''} = \frac{\Delta T_s}{\Delta d} \, \sin(\delta), \tag{2}$$

where $A_r$ is the area of the torque-displacement loop, $\delta$ is the phase angle, $T_s$ is the torque, and $d$ is the lateral displacement, following the definition in [52].

Fig 8(a) shows the experimental setup for the oscillatory shear test for the viscoelastic properties of a thrombus sample [53], where the diameter of the rheometer disk is 50 $mm$ and the gap width is 300 $\mu m$. Fig 8(b) and 8(c) show the computational domain, which is a 3D channel with $\Omega = \{(x, y, z)\vert \, 0 \leq x \leq 6, 0 \leq y \leq 1, 0 \leq z \leq 1\}$. The upper plate moves at a sinusoidal velocity $v = 0.2 \sin(1.2\pi t)$, with a periodic boundary condition for velocity in the $z$-direction

**Table 2. Parameters used for oscillatory shear simulations in dimensional units.**

|  | Frequency (Hz) | Velocity (m/s) | $\frac{\rho_2}{\rho_1}$ | $\frac{\eta_2}{\eta_1}$ | $\lambda_e$ (Pa) |
|---|---|---|---|---|---|
| case1 | 0.5 | 6.67e-4 | 2 | 2 | 0.22 |
| case2 | 0.5 | 6.67e-4 | 2 | 2 | 0.44 |
| case3 | 0.5 | 6.67e-4 | 2 | 2 | 0.67 |
| case4 | 0.5 | 6.67e-4 | 2 | 2 | 0.89 |

and a no-slip boundary condition on the lower plate. In our simulations, the characteristic velocity and length scales (required for non-dimensionalization of the equations) are $3.33 \times 10^{-3}$ $m/s$ and $3 \times 10^{-4}$ $m$, respectively. Zero-Neumann boundaries are imposed for the phase-field variable and for the deformation related variable $\psi$ in 2D, specifically, we impose $\mathbf{n} \cdot \nabla \phi = 0$, $\mathbf{n} \cdot \nabla(\Delta \phi) = 0$, and $\mathbf{n} \cdot \nabla \psi = 0$. The thrombus is considered to be in the middle of the channel and three different initial volume fractions VF = 0.3419, 0.5129, and 0.6335 are examined. At the frequency of 0.5 $Hz$, the velocity of the upper plate is estimated to be $6.67 \times 10^{-4}$ $m/s$ with the fluid kinematic viscosity $\eta_1 = 1e\text{-}6$ $Pa \cdot s$ and density $\rho_1 = 1000$ $kg/m^3$. We compute the displacement at the thrombus interface, and the shear stress (including the viscous and the elastic contributions) acting on the surface where the thrombus contacts the upper plate. All parameters for the simulations are listed in Table 2.

We construct the stress-displacement loops as shown in Fig 8(d)–8(k). For the same volume fraction shown in each row, the shear stress increases as the elastic shear modulus increases. Moreover, for the same elastic shear modulus on each column, the shear stress increases with increasing volume fraction. Our simulation results in Fig 9 show both $G'$ and $G''$ increase as the concentration of fibrinogen increases.

Fig 10 shows the results of calibrating the relaxation time for the rheometry experiment. The dotted green line shows the relaxation time computed based on experiments corresponding to initial fibrinogen concentrations of 1, 3 and 6 $mg/mL$, where $G' = 10^{2.6*log_{10}(c_{Fbg})+log_{10}(10)}$. In addition, we estimate $G'' = 10^{1.7*log_{10}(c_{Fbg})+log_{10}(0.7)}$, and the relaxation time is calculated using $\lambda_s = [G''/(2\pi f) - \eta_1]/G'$ [51]. The solid lines show the relaxation time from extrapolation and extension from our simulations, where we use a linear fit for the simulation data (only the two points corresponding to VF = 0.3419 and 0.5219) and extrapolate to the concentration range of 1 − 6 $mg/mL$. By comparing our simulation results with experimental data (see Fig 10), we select $\lambda_e = 0.44$ Pa. As a result, we set the relaxation time as $\lambda_s(\phi) = 10^{-0.7172*log_{10}(c_{Fbg})-1.1140}$ in the following 3D simulation.

## Simulation of thrombus formation and deformation in an idealized aneurysm

In the case of an abdominal aortic aneurysm (AAA), the low shear rate region inside the most enlarged region combined with a long shear history experienced by the entering platelets can promote activation and aggregation of the platelets and thus thrombus initiation [54, 55]. Thrombus, in turn, causes further enlargement of the AAA and possible thromboembolic events [56, 57]. To demonstrate the capability of the proposed framework in predicting thrombus deformation and possible embolism, in this section, we model the initiation and development of a thrombus in an idealized AAA as well as the subsequent interaction between the thrombus and blood flow.

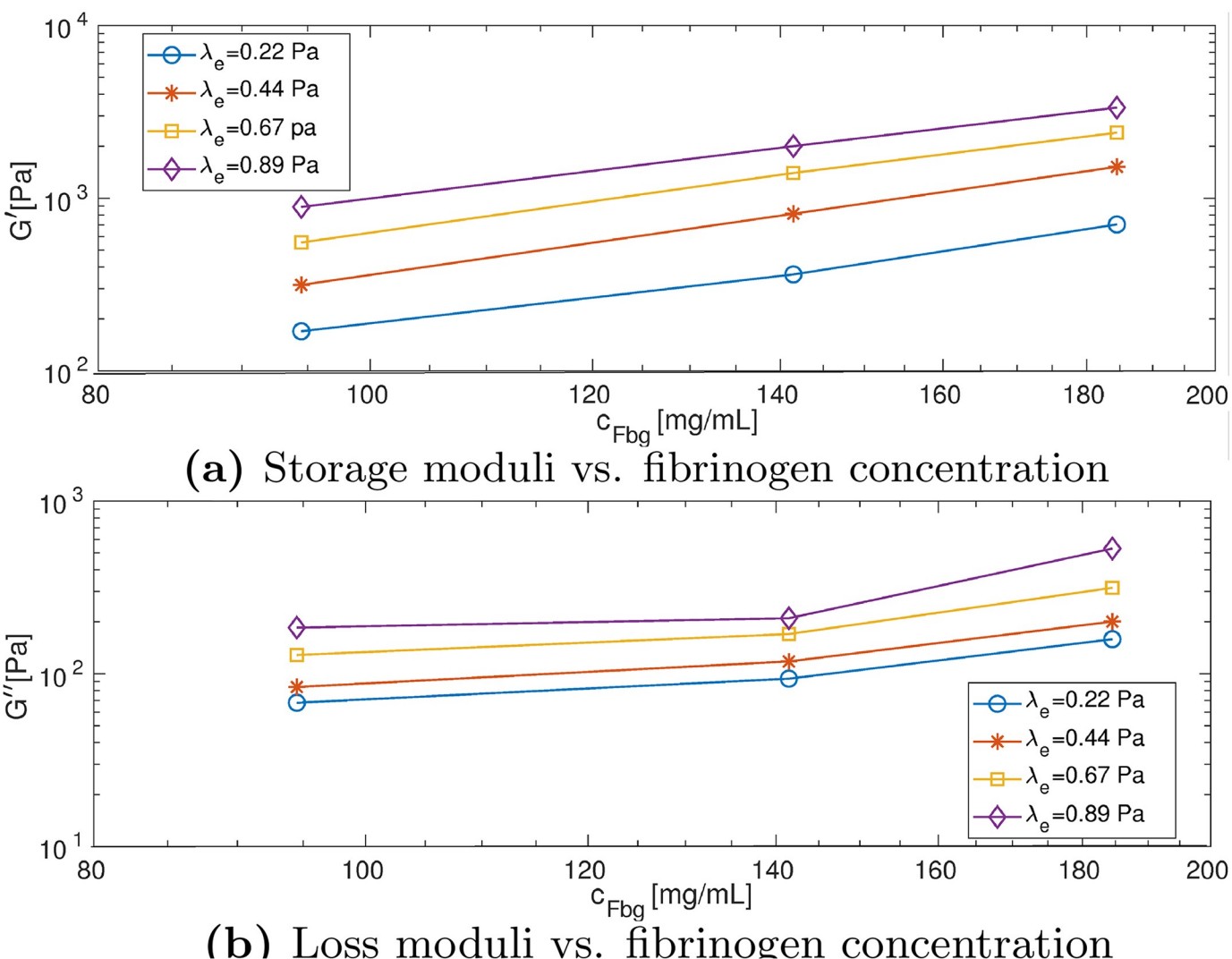

**Fig 9. Estimating viscoelastic properties of thrombus using the phase-field model.** (a) Storage ($G'$) and (b) loss moduli ($G''$) as a function of fibrinogen concentration $c_{Fbg}$ at different $\lambda_e$ computed from simulation results.

## Modeling thrombus formation with FCM

To avoid solving directly the spatio-temporal "thrombus deposition potential" [54, 55], we compare three realistic platelet deposition sites for illustrative purposes, namely, on the proximal and distal neck and the bottom of the aneurysm, respectively, as shown by the blue patches in Fig 11. Initially, passive platelets (particles) are distributed uniformly inside the aneurysm (Fig 11(a)). When the passive platelets interact with the deposition sites, they become triggered and change to an activated state after a delay time $\tau_{act} = 0.1 - 0.3s$ [33, 48]. Upon activation, the platelets begin to adhere at deposition sites and to other activated platelets, then modeled as pseudo-platelets (*i.e.*, platelets and associated fibrin) growing in size to an effective radius $r_{eff}$ (as illustrated by the schematic in Fig 11(b)). This coarse-grained approach reduces the computational cost by simulating fewer platelets than the physiological concentration, following our prior work [3]. The Reynolds number of the flow is $Re = 181.9$, which is calculated

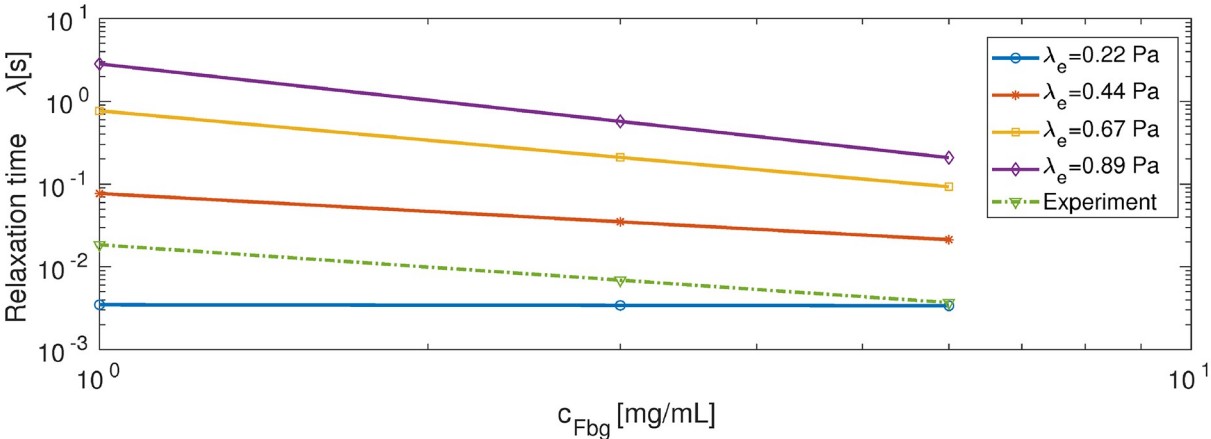

**Fig 10. Relaxation time as a function of fibrinogen concentration $c_{Fbg}$ corresponding to different $\lambda_e$.** Simulation results are represented by solid lines (data are fitted and extrapolated) and experimental result is represented by the green dotted line [51].

based on blood viscosity $3.77 \times 10^{-3} \, Pa \cdot s$ and blood density $1060 \, kg/m^3$. A generic pulsatile blood flow is prescribed at the inlet for which the average velocity profile within one cardiac cycle is shown in Fig 11(c).

Once the simulation begins, the platelets close to the deposition sites become activated and adhered to the wall. As the simulation continues, increased numbers of activated platelets adhere to the aggregates. We find that platelet aggregates become stable after simulating 15 cardiac cycles. As shown in Fig 12(a), platelet aggregates are mainly located around the platelet deposition sites at the neck and bottom of the aneurysm, consistent with findings from prior computational modeling [54]. Next, as illustrated in Fig 12(b), we convert the Lagrangian distribution of the platelet aggregates to volume fractions that can be used as an initial condition for the subsequent phase-field modeling of interactions between the thrombus and blood flow with possible embolization.

## Multiphase continuum modeling of a thrombus

After the initial phase of platelet aggregation modeled by FCM, we simulate the interaction of blood and thrombus under the same flow conditions using the phase-field model. We assume that platelets will no longer accumulate on the thrombus or wall. The permeability $\kappa(\phi)$ calculated by the Davies' equation and the thrombus viscoelastic properties $\lambda_e(\phi)$ estimated in Results section are used to represent the biomechanical properties of the thrombus.

Fig 13 shows a sequence of snapshots of the thrombus shape under the pulsatile flow in different cross-sectional planes, where the detachment of small pieces of thrombus due to the interaction with blood flow is observed at different axial locations. There is mild pinch-off at both the upstream $x \approx 5cm$ and downstream $x \approx 8.5$ cm of the thrombus, while moderate thrombus contraction occurs proximal to the aneurysm at $x \approx 5.5$ cm, 6.5 cm, and 7.5 cm. Thrombus volume fraction VF is low close to the center of the vessel and distal to the aneurysm higher in the deeper regions of the aneurysm. We also perform simulations with a steady inflow boundary condition to investigate the effect of pulsatility on the results. As shown in Fig 6 in Supporting information, our simulations show that a thrombus under steady flow does not release emboli, which is different from the case of pulsatile flow shown in Fig 13. This difference suggests that pulsatility of the blood flow could be important in thromboembolic events secondary to intraluminal thrombus in AAAs.

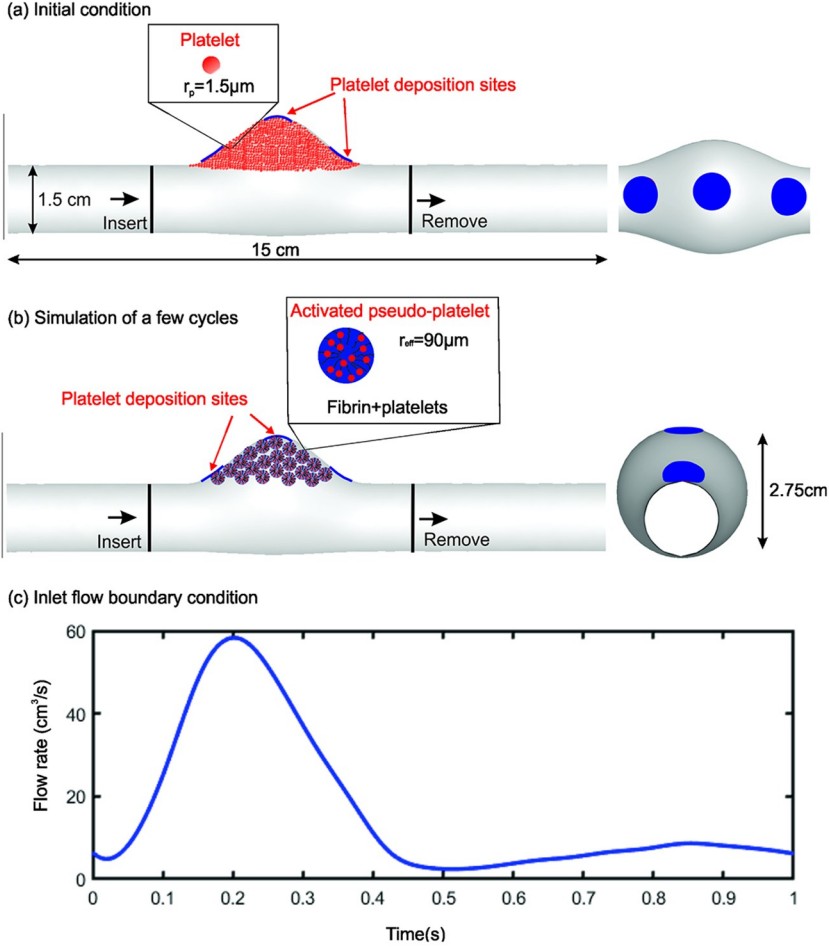

**Fig 11. Geometry and boundary condition for simulating thrombus formation in a fusiform aortic aneurysm.** (a), (b) Passive platelet particles are initially distributed throughout the computational domain, including within the aneurysm, while new platelets are inserted into the proximal domain and outgoing platelets are deleted from distal domain. *Upper inset*: passive platelets as spherical particles of radius $r_p$ = 1.5 $\mu m$. Pseudo-platelets change their size once they come into contact with the deposition sites and become activated, catalyzing the conversion of blood borne fibrinogen into semi-solid fibrin. *Lower inset*: activated platelets with increased effective radius $r_{eff}$ = 60$r_p$. (c) Pulsatile flow profile imposed at the inlet; a zero pressure condition is prescribed at the outlet.

## Discussion

In this work, we present a multiscale framework that combines FCM with a phase-field method to model both the initial formation of a thrombus through platelet aggregation and fibrin deposition and subsequent interactions between the flowing blood and the thrombus. Such modeling is challenging because of the multiphysics and multiscale biomechanics. Neither the popular ALE approach for fluid-structure interactions nor the level-set method have been applied to model these aspects of a thrombus due to the difficulty in capturing the complicated rheology at the interface of the thrombus and blood flow. The phase-field method, on the other hand, can capture these effects naturally because it is fully-Eulerian and able to capture the interfacial rheology and the shear stress on the thrombus surface. A recent numerical study [31] simulated thrombus deformation with a 2D phase-field model, but did not consider how the viscoelastic properties of the thrombus affect its deformation. In this work, we implemented a 3D phase-field model to investigate effects of the permeability and viscoelastic

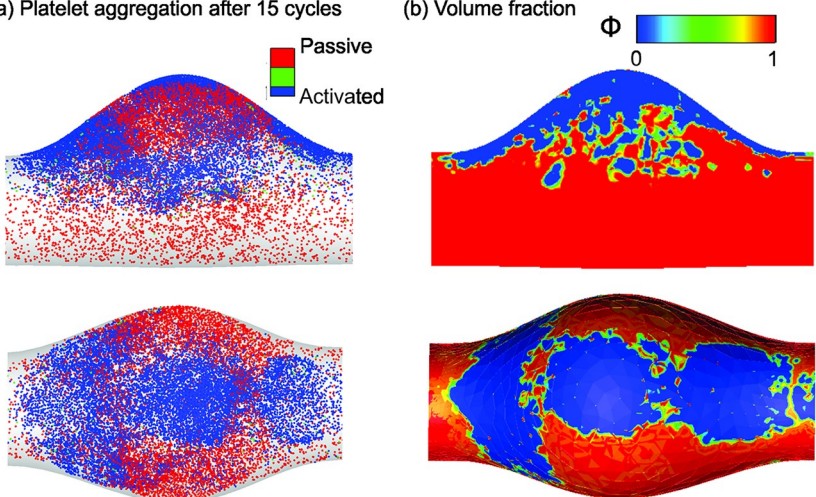

**Fig 12. Platelet aggregation modeled by the FCM and the initial conditions for the phase-field model.** (a) Side and top views of the simulated platelet aggregates in the aneurysm after 15 cardiac cycles. (b) The corresponding input for the phase-field simulation, where the top row is the central cross-section of the vessel and the bottom row is the view from the top.

properties of a thrombus on its deformation in a flow field. Our simulations show that the permeability and the elastic shear modulus are essential in determining the responses of the thrombus. Our simulations of the rheology and flow-induced shape changes of the thrombus were consistent with results in a prior computational study [31] and experimental observations [29, 50], suggesting the potential of this approach in predicting the risk of thrombotic and thromboembolic events under physiologic and pathological conditions. Indeed, even though we used the volume fraction of a thrombus to define the phase-field variable $\phi$, which leads to a finite thickness interface in our simulations that may compromise the original phase-field assumption of a thin smooth transition layer, our assumption simplified the model and yielded a reasonable congruence with available additional data [49].

A unique feature of the proposed framework is the combination of a phase-field method with FCM by converting the volume fraction of platelet aggregation into a phase-field variable, which enables consistent modeling of thrombus initiation, development, and deformation under blood flow. When simulating thrombus formation in an idealized aortic aneurysm, our FCM simulations suggest that the thrombus does not grow uniformly; moreover, the volume fractions of the blood and thrombus also vary spatially. Portions of thrombus with lower volume fractions VF have a larger propensity to deform toward the downstream direction of the blood flow. Small pieces even can detach from the bulk thrombus and flow downstream, similar to the process of thrombus embolization.

We note that thrombus can form via different mechanisms that lead to different compositions and stability [58, 59]. The composition of thrombus in AAAs is even more complex. Intraluminal thrombus may contain layers with different compositions, representing blood cells that were captured over multiple time scales [60]. Complexities similarly arise in intramural thrombus in the aortic dissection, with some regions demonstrating remodeling of fibrin-rich thrombus into collagen-rich fibrotic tissue [21]. In this work, we considered the formation of a thrombus in an idealized AAA, in which the low shear rate region inside the aneurysm combines with a long shear history experienced by the platelets to promote local platelet deposition [54, 55]. The aim of our simulations was thus not to predict the differential formation of

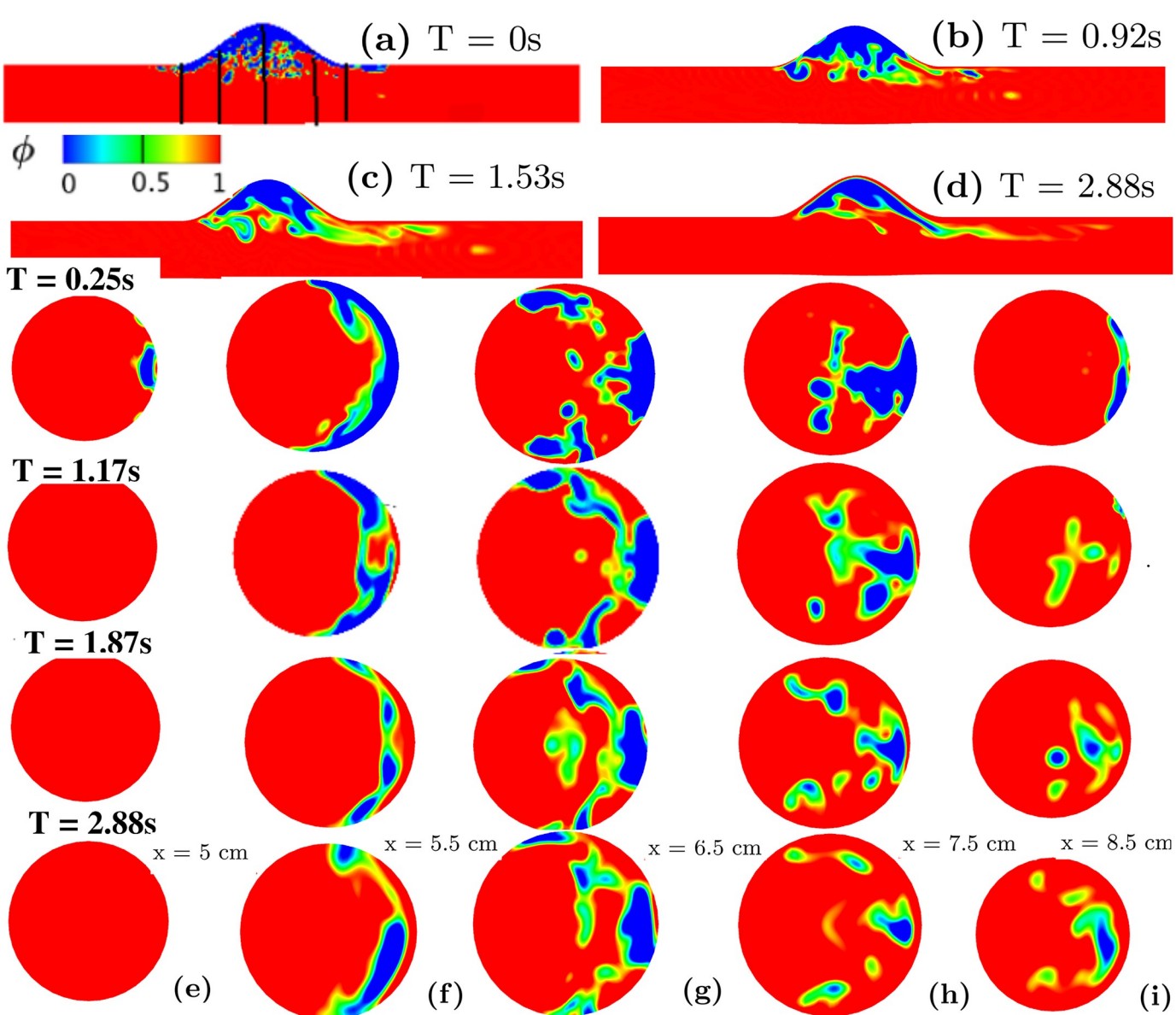

**Fig 13. Physiologic phase-field modeling of thrombus deformation in an idealized aneurysm.** Contours of thrombus volume fraction in the aneurysm shown using the phase-field variable at (a) $T = 0s$ (black lines indicate planar cross-sections at $x = 5, 5.5, 6.5, 7.5,$ and 8.5 cm), (b) $T = 0.92s$, (c) $T = 1.53s$, and (d) $T = 2.88s$. Rows 3-6: phase-field contours plotted at different $x$ locations (e) x = 5cm (f) x = 5.5 cm (g) x = 6.5 cm (h) x = 7.5 cm (i) x = 8.5 cm varying with time. Third row: $T = 0.25s$; Fourth row: $T = 1.17s$; Fifth row: $T = 1.87s$; Sixth row: $T = 2.88s$.

a thrombus in aneurysms, but rather to demonstrate the capability of the proposed framework to simulate the formation of thrombus that initiated via platelet activation and deforms due to both external and interstitial blood flow. Finally, we calibrated both the permeability and viscoelasticity of the model using data from fibrin gels given the lack of data on the viscoelastic properties of platelet-rich thrombus. These parameter values can be refined when new experimental data become available.

Taken together, we developed a 3D phase-field model for simulating blood-thrombus interactions. Calibration of model parameters was achieved by comparing the simulation results

with existing *in vitro* experimental data for the poro-viscoelastic behavior. The proposed 3D phase-field model can be combined with FCM to simulate the process from platelet activation and deposition to subsequent deformation while exposed to steady or pulsatile blood flow. Guided by patient-specific clinical data, such as lesion geometry and the local blood flow rate, this multiscale framework has the potential to predict the risk of thrombotic-embolic events, which are responsible for significant morbidity and mortality. Furthermore, this framework can be further expanded by combining it with models for thrombus remodeling [61–63] to improve our understanding of how changes of biomechanical properties of a thrombus during its maturation affect its role in both physiology and pathophysiology.

## Methods

### Platelet transport and aggregation

We simulate the transport of platelets within flowing blood and their aggregation on deposition sites using FCM integrated with SEM [33]. FCM is used to describe the motion of platelets and their (bi-directional) interactions with the flowing blood, which is determined using SEM to solve the flow field on a fixed Eulerian grid. This multiscale approach has been successfully used in modeling platelet aggregation in venules [48], stenotic channels [33], and intramural aortic dissections [3]. In our simulations, platelets exhibit three different states, namely *passive*, *triggered*, or *activated*. In passive or triggered states, platelets have their physiological radius $r_p = 1.5 \ \mu m$ and are non-adhesive. Once platelets in the passive state interact with activated platelets or a deposition site, they switch to a triggered state and then activate after a delay time $\tau_{act} = 0.1 - 0.3 \ s$ [48]. After activation, pseudo-platelets of platelets embedded in a local fibrin network) grow to an effective radius $r_{eff} \sim 60 r_p$. This approach allows us to use fewer platelets than the physiological concentration when modeling thrombus formation in large computational domains [3].

The margination of platelets due to collisions with blood cells in flowing blood has been studied extensively, both experimentally and theoretically, for straight channels and idealized vessels [64–68]. To avoid the added complexity of simulating blood cells explicitly in the blood flow, we account for platelet margination by using a master profile for platelet distribution within the aorta, quantified in in [64] when platelet particles are inserted into the inlet of a vessel. The interactions amongst the platelets as well as between the platelets and the deposition sites are then described by a Morse potential (attractive interactions between platelets) and an exponential repulsion potential (exclusive effects of the platelet particles) (see S5 Text). The adhered platelet particles are considered to be 'stationary' or part of the thrombus when their moving distance within one cardiac cycle is less than 1/100 of their diameter. The interaction forces are shear-rate dependent; the force values were calibrated using data from four independent studies, including two *in vivo* [3, 27] and two *in vitro* [69, 70] experiments, which measured platelet aggregation at different shear rates. More detailed information of the FCM model can be found in S5 Text and in [3, 33].

To compute the volume fraction of the formed thrombus *VF* (i.e., the local volume occupied by the thrombus divided by the volume of blood), we first use a continuum representation of the adhered platelets from FCM simulations to evaluate the local volume fraction of pseudo-platelet particles as $\Psi_{fcm}(\mathbf{x}, t) = \sum_{n=1}^{N} \mathcal{V}_p^n \Delta(\mathbf{x} - \mathbf{Y}^n(t))$, where $\mathcal{V}_p^n$ is the volume of each pseudo-platelet, $\mathbf{x}$ is the position of the background Eulerian grid, and $\mathbf{Y}^n$ is the position of each FCM particle. Knowing the local concentration of fibrinogen and platelets and using the computed $\Psi_{fcm}$ field, we estimate the thrombus volume fraction as $VF = [\Psi_f(c_{Fbg}) + \Psi_p(c_{plat})] \Psi_{fcm}$, where $\Psi_f$ is the volume fraction of fibrin and $\Psi_p$ is the volume fraction contribution of platelets [3, 49]. $\Psi_f$ can be computed by an empirical relation $\Psi_f = c_{Fbg}/[\rho_{Fbg}(0.015 \log(c_{Fbg}) +$

0.13)], where $c_{Fbg}$ ($mg/mL$) is the concentration of fibrinogen and $\rho_{Fbg} = 1.4$ $g/mL$ is the density of a single fibrinogen molecule [49]. $\Psi_p$ can be evaluated directly based on each platelet's volume and the local platelet number density [49].

## Modeling clot deformation using the phase-field method

Various computational methods have been implemented to model interactions between fluids and viscoelastic solids, including the arbitrary Lagrangian Eulerian formulation (ALE), and the immersed boundary method as well as, level set, volume-of-fluid, and phase-field methods [71–73]. An advantage of phase-field approaches in modeling multiphase systems is that they describe the system by characterizing the free energy. In this way, different physical/biological phenomena can be considered through an appropriate modification of a unified set of governing equations for the free energy formulated over the entire computational domain [71]. The phase-field method has been used to address interactions between flowing blood and thrombus as well as between the flowing blood and a biofilm [31, 38]. These studies were limited, however, to 2D simulations for predefined thrombus and biofilm shapes. Other studies elucidated the roles of thrombus permeability on its deformation and embolization [31], but did not include effects of spatial variability of the volume fraction of the thrombus. In the current study, we extend the prior work by performing new 3D simulations of interactions between the thrombus and flowing blood, while considering both the spatially variable permeability and viscoelastic properties of a thrombus. We believe that this approach provides more biologically realistic and physically consistent results.

The key idea of the phase-field approach is to use a phase-field variable $\phi$ to describe a two-phase system, with the interface between two different phases modeled by a thin smooth transition layer [71]. In our simulation of a vascular lumen $\Omega$ containing a thrombus, $\phi = 1$ represents the blood, $0 < \phi < 1$ represents a mixture of blood and thrombus, and $\phi = 0$ represents thrombus only. The phase-field equation is derived by minimizing the total energy $E_{tot}$ of the blood-thrombus system, which is the sum of the kinetic energy, cohesive energy of the mixture, and the elastic energy of the fibrin network and platelets in a thrombus. The cohesive energy $E_{cohesive}$ of the mixture is adopted from Cahn & Hilliard [74]. We describe the viscoelastic fibrin network in thrombus using a Kelvin-Voigt type model where a neoHookean relation is used to describe the elastic behavior and a dashpot is used to describe the viscous behavior [51, 75].

The total free energy for the blood-thrombus system can thus be written by adding the cohesive and elastic energies, respectively, as

$$E_{tot} = \int_\Omega [\frac{\lambda}{2}|\nabla\phi|^2 + \frac{\lambda}{2h^2}\phi^2(\phi-1)^2]d\mathbf{x} + \int_\Omega [\frac{\lambda_e}{2} tr(\mathbf{F}^T\mathbf{F} - \mathbf{I})]d\mathbf{x}, \qquad (3)$$

where $\lambda$ is the mixing energy density, $\lambda_e$ is the elastic shear modulus, $h$ is a characteristic length scale of the interface thickness, $tr$ is the trace of a matrix and $\mathbf{I}$ is the identity matrix. Relative to an Eulerian reference frame, the deformation gradient tensor $\mathbf{F}(\mathbf{x}, t) = \partial\hat{\mathbf{x}}/\partial\mathbf{X}$ satisfies the following evolution equation [75] (it can be shown that $\nabla \cdot \mathbf{F} = \mathbf{0}$ if $\nabla \cdot \mathbf{F}_0 = \mathbf{0}$ at time $t = 0$):

$$\frac{\partial\mathbf{F}}{\partial t} + \mathbf{u} \cdot \nabla\mathbf{F} = \nabla\mathbf{u} \cdot \mathbf{F}, \qquad (4)$$

where $\mathbf{u}$ is the velocity, $\hat{\mathbf{x}}$ and $\mathbf{X}$ are the current and original locations of a material point. In prior 2D models [31, 38], an auxiliary vector $\psi$ was introduced to calculate the deformation gradient tensor $\mathbf{F}$ through $\mathbf{F} = \nabla \times \psi$ and the corresponding equation for $\psi$ is $\frac{\partial\psi}{\partial t} + \mathbf{u} \cdot \nabla\psi = 0$.

To improve the computational efficiency in 2D, we solve for $\psi = [\psi_1, \psi_2]$ instead of the four components of Eq 1 in Supporting information.

The governing equations for the 3D phase-field model are summarized as follows

$$\rho(\frac{\partial \mathbf{u}}{\partial t} + \mathbf{u} \cdot \nabla \mathbf{u}) + \nabla p - \nabla \cdot (\eta(\phi)\nabla u) = $$

$$\nabla \cdot (\lambda_e (1 - \phi)(\mathbf{F}\mathbf{F}^T - \mathbf{I})) - \lambda \nabla \cdot (\nabla \phi \otimes \nabla \phi) - \eta(\phi)\frac{(1 - \phi)\mathbf{u}}{\kappa(\phi)}, \tag{5a}$$

$$\nabla \cdot \mathbf{u} = 0, \tag{5b}$$

$$\frac{\partial \mathbf{F}}{\partial t} + \mathbf{u} \cdot \nabla \mathbf{F} = \nabla \mathbf{u} \cdot \mathbf{F} \tag{5c}$$

$$\frac{\partial \phi}{\partial t} + \mathbf{u} \cdot \nabla \phi - \tau \Delta \mu_1 = 0, \tag{5d}$$

$$\mu_1 = \frac{dE_{tot}}{d\phi} = -\lambda \Delta \phi + \lambda \gamma g_1(\phi) + \frac{\lambda_e}{2} tr(\mathbf{F}^T\mathbf{F} - \mathbf{I}), \tag{5e}$$

where $u, p(\phi), \rho(\phi), \eta(\phi), \kappa(\phi)$ are the velocity, pressure, mass density, dynamic viscosity and permeability, respectively, with the phase-field variable $\phi \in [0, 1]$. Furthermore, $\gamma$ is a constant interfacial mobility of the phase-field, $\tau$ is a relaxation parameter, and $g_1(\phi)$ is the derivative of the double well potential $\phi^2(\phi - 1)^2/2h^2$. Eq 5(a) and 5(b) account for linear momentum balance and mass balance, respectively. Eq 5(c) calculates the deformation gradient tensor $\mathbf{F}$ and Eq 5(d) computes the phase-field variable $\phi$. The free energy potential $\mu_1$ for the blood-thrombus system is expressed by Eq 5(e), which is derived by taking the derivative of the total free energy $E_{total}$ with respect to $\phi$.

Following Dong *et al.* [71], our model and algorithms have several attractive features. The governing equations are solved on the entire domain with a fixed Eulerian grid and all key variables, such as the velocity $\mathbf{u}$, the phase-field $\phi$ and the deformation gradient tensor $\mathbf{F}$ are decoupled. The resulting coefficient mass and stiffness matrices are constant and thus can be computed at the beginning of each simulation to increase the computational efficiency. Furthermore, the $4^{th}$ order phase-field equation (*i.e.*, Cahn-Hilliard equation [74]) is reformulated into two second-order Helmholtz equations, which can be solved by the spectral or finite element method. We use an entropy viscosity method to stabilize the hyperbolic Eq 5(c) for $\mathbf{F}$ (or vector $\psi$ in 2D) [76].

The surface tension parameter $\sigma$ is related to the mixing energy density $\lambda$ in Eq 5 using $\lambda = \frac{3}{2\sqrt{2}}\sigma h$. We tested different values of the surface tension ($\lambda$ changes accordingly) with other parameters fixed, and found that the effect of $\sigma$ on the mechanical properties of the thrombus is negligible. Therefore, we assume a weak surface tension force in the momentum Eq 5(a) compared to the viscous and elastic forces.

## Supporting information

**S1 Text. Governing equations in 2D.**
(PDF)

**S2 Text. Thrombus permeability and deformation tests in a 2D channel.**
(PDF)

**S3 Text. Sensitivity tests of interface width & surface tension.**
(PDF)

**S4 Text. Simulation of thrombus deformation under steady flow.**
(PDF)

**S5 Text. Modeling platelet transport and aggregation using FCM.**
(PDF)

**S1 Fig. Permeability test in a 2D channel.** Stream-wise velocity field $u$ for case 3 (see Table 1) with $h_s = 1.4$ and $h_c = 0.6$ at (a) $\kappa_s = 1$, (b) 0.1, (c) 0.01, and (d) 0.005 shown at time $T = 0.7$.
(TIFF)

**S2 Fig. Shear stress acting on the clot surface in a 2D channel.** Shear stress (normalized by the stress $F_0$ calculated with $\kappa_s = 5e - 3$) acting on the surface of the thrombus as a function of $\kappa_s$, with case 2 (black line) and case 3 (line) corresponding to different sizes of shell subdomain of the thrombus.
(TIFF)

**S3 Fig. Thrombus shapes in a 2D channel at different flow velocities.** Impact of the flow velocity $u$ on thrombus deformation with $h_s = 0.6$ and $h_c = 0.3$. Phase-field contour of the thrombus at different shear rates are plotted for (a) $u = 0.2$, (b) 0.5, (c) 1.0 and (d) 2.0 at time $T = 0.48$. $u$ is the maximum velocity at the inlet.
(TIFF)

**S4 Fig. Sensitivity of interface width on the deformation of thrombus in a 2D channel.** $1^{st}$ row: axial velocity and $2^{nd}$ row: phase-field profile at $T = 1$. h is the interface width.
(TIFF)

**S5 Fig. Sensitivity of surface tension on the deformation of blood clot in a 2D channel.** Phase-field profile at $T = 1$. $\sigma$ is the surface tension.
(TIFF)

**S6 Fig. Thrombus deformation under a steady flow.**
(TIFF)

## Author Contributions

**Conceptualization:** Xiaoning Zheng, Jay D. Humphrey, George E. Karniadakis.

**Data curation:** Xiaoning Zheng.

**Formal analysis:** Xiaoning Zheng.

**Funding acquisition:** Jay D. Humphrey, George E. Karniadakis.

**Investigation:** Xiaoning Zheng.

**Methodology:** Xiaoning Zheng, Alireza Yazdani.

**Software:** Xiaoning Zheng, Alireza Yazdani.

**Supervision:** Jay D. Humphrey, George E. Karniadakis.

**Validation:** Xiaoning Zheng, Alireza Yazdani.

**Visualization:** Xiaoning Zheng, Alireza Yazdani, He Li.

**Writing – original draft:** Xiaoning Zheng, Alireza Yazdani, He Li, Jay D. Humphrey.

**Writing – review & editing:** Xiaoning Zheng, Alireza Yazdani, He Li, Jay D. Humphrey, George E. Karniadakis.

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
