## [Decision Letter · Decision Letter 0]

18 Oct 2019

Dear Dr Karniadakis,

Thank you very much for submitting your manuscript 'A Three-dimensional Phase-field Model for Multiscale Modeling of Thrombus Biomechanics in Blood Vessels' for review by PLOS Computational Biology. Your manuscript has been fully evaluated by the PLOS Computational Biology editorial team and in this case also by independent peer reviewers. The reviewers appreciated the attention to an important problem, but raised some substantial concerns about the manuscript as it currently stands. While your manuscript cannot be accepted in its present form, we are willing to consider a revised version in which the issues raised by the reviewers have been adequately addressed. We cannot, of course, promise publication at that time.

Sincerely,

Alison Marsden

Associate Editor

PLOS Computational Biology

Jason Haugh

Deputy Editor

PLOS Computational Biology

[LINK]

Reviewer's Responses to Questions

**Comments to the Authors:**

Reviewer #1: Attached

Reviewer #2: The authors present a framework to integrate the Force Coupling Method (as a means to model platelet transport in fluid) and a Phase-field modeling technique (as a means to model a thrombus as a poro-elastic structure) to investigate thrombus formation and deformation. To calibrate parameters within their phase-field model, they compare to data from in vitro experiments of clot permeability and rheology. The goal is to use this framework in the future to help determine risk of thrombotic events like embolization in patient-specific geometries. The framework presented is an extension from their recent paper and adds phase field models developed by a different group. The ideas in this paper are very interesting and suited for the journal, but there are many details missing related to the platelet aggregation portion of the model and so that section of the paper is difficult to fully and fairly review. Further, the calibrations for permeability and rheology were based on studies with fibrin rich clots instead of platelet rich clots, which leaves doubt about the relevance for cell-rich clots. See detailed comments below.

Major comments:

1. It is unclear to the reviewer how the platelets (modeled by FCM) are adhering and aggregating, and how their “growth” is considered dynamically in time and in the fluid. In section 2.1 (platelet transport and aggregation) there is no description of how platelets aggregate or adhere to the wall. At what point do they begin to “grow”? How is the spatial distribution of the new larger platelets determined from the distribution of the smaller platelets, especially near the wall? Are the larger platelets stationary? Finally, during the growth stage, how do the authors update the fluid equations surrounding the platelets? Due to lack of details provided on the thrombus formation process makes difficult to fully assess and properly review the study.

2. The usefulness of going from FCM to a PF model is the heterogeneity of the clot – then the effects of heterogeneity on the mechanics of embolization can be studied; the structure of the thrombus that initially forms is clearly extremely important. Without describing how the platelets are aggregating and which ones have actually aggregated, it is not clear that the phase field model is a correct representation of the clot itself, but rather just a mass of platelets that are flowing through the fluid, in a non-aggregated state. It is possible that this is distinguished somehow but it is not made clear to the reviewer.

3. It is unclear to the reviewer where fibrinogen fits into this model. In the methods section, the concentration of fibrinogen is mentioned, and then again in the rheological study section it is used, but there is no evolution equation for fibrinogen for it to exist or interact in the thrombus or at the wall for adhesion.

4. The function g1(phi) represents a double well potential when phi is in [-1,1] with minima at +/-1; what is the rationale for choosing phi in [0,1] in this model?

5. The calibration of the permeability of the thrombi were based on experiments with fibrin clots and the empirical formula of Davies is based on fibrous material. The referenced paper (Wufsus et al.) included studies on platelet rich clots where the permeabilities were fit well with a different empirical formula (Ethier). This suggests that the permeabilities of the thrombi modeled in the current study could be significantly underestimated. The authors should instead fit to the platelet-rich clot data or justify their choice for using the fibrin clots instead of platelet-rich ones.

6. Similarly, the calibrations for the shear modulus are based on rheology experiments performed with fibrin gels while the clots in this study are assumed to be primarily platelets. The section describing how the volume fraction is calculated based on fibrinogen concentration is very confusing. It is not clear what the parameters are in equation 5 or how they relate to the ‘VF’. The final sentence in that section relates the relaxation time to a fibrinogen concentration but there is no fibrinogen in the model. This section should be rewritten more carefully and explicitly state what the outcome is.

7. What is the bases for having the AA completely filled with platelets as the initial placement? It seems that one of the benefits of modeling the initial thrombus formation using FCM is that it will lead to more interesting and physiologically relevant thrombus formations. To keep inline with the journal criteria for significant biological insight, it would enhance this study to show a simulation with clots that have grown in the AA that initially was empty.

8. The claim is that the timescale of the phase-field modeling covers hours of time (Figure 1), but this claim is not justified by the simulations provided in this paper. Can the authors comment on this?

Minor comments:

1. The sensitivity of the surface tension parameter was tested for fixed values of h (and other parameters). The sensitivity was said to be negligible, but it was not clear what tests were used to come to that conclusion. Is the parameter sensitive as h changes? I don’t believe h was ever specified either.

2. In the clot deformation in 2D/3D section, how thick was the z-direction and were the results sensitive to that? As for the two densities and viscosities, are those for inside and outside the thrombus and which is where?

3. “calibration” would be a better description than “validation” for the permeability model, as it is truly calibration of a model representation.

4. Units. In most of the results sections, the units were left off of the parameters. It would be helpful to at least see units on the permeability

5. In the Modeling thrombin formation with FCM section, the term deposition sites is used, but in the figure, initiation sites is used. Also the reference to figure 11b on the last line of the first paragraph should reference 11c

6. What is the upstream distribution of platelets during the dynamic simulation?

**Have all data underlying the figures and results presented in the manuscript been provided?**

Reviewer #1: Yes

Reviewer #2: Yes

PLOS authors have the option to publish the peer review history of their article (what does this mean?). If published, this will include your full peer review and any attached files.

Reviewer #1: No

Reviewer #2: No

---

## [Decision Letter · Decision Letter 1]

3 Feb 2020

Dear Professor Karniadakis,

We are pleased to inform you that your manuscript 'A Three-dimensional Phase-field Model for Multiscale Modeling of Thrombus Biomechanics in Blood Vessels' has been provisionally accepted for publication in PLOS Computational Biology.

Before your manuscript can be formally accepted you will need to complete some formatting changes, which you will receive in a follow up email. A member of our team will be in touch within two working days with a set of requests.

Reviewer #1's constructive comments should also be considered, and the text revised where appropriate.

Best regards,

Alison Marsden

Associate Editor

PLOS Computational Biology

Jason Haugh

Deputy Editor

PLOS Computational Biology

Reviewer's Responses to Questions

**Comments to the Authors:**

Reviewer #1: The revision has been improved considerably. However, there are further comments that should be addressed before publication.

Multiple places: The authors claim white clot is primarily made of fibrin and platelets, which is not consistent with early (Cadroy 89) and recent descriptions (e.g. Jackson 07 or Kim 19). White clot is composed of vWF and platelet. Fibrin is in red clots that may form on top of white clots after the occlusion causes stagnation of the blood.

Question 10: It was understood that the phase-field parameter ϕ is defined in terms of volume fractions. However, the authors use ϕ = 1 to denote blood flow and ϕ = 0 for thrombus. This is not consistent with the custom that the authors use for ϕ_f to denote the volume fraction of fibrin (solid), etc. I suggest defining ϕ = 0 as blood flow and thrombi as ϕ = 1 in the phase field model.

Question 12: the authors validate the permeability calculation using existing fibrin gel data. Fibrin gel is not a thrombus. It is OK to validate the model this way given the limited experimental data. However, the author should not oversell it by calling it "validation of the permeability of the thrombus". This is only a validation of the permeability calculation using fibrin gel data.

Question 13: It is understood that eventually an equilibrium configuration of the clot forms. However, this equilibrium is dynamic, meaning there is thrombi formation and rupture (or platelet adhering and detaching). This answer still does not address the fact the platelet aggregation may keep forming even during so-called "reconfiguration" process when emboli occurs. I suggest the authors state that a simplification/assumption is that they neglect the further formation of thrombi during the phase-field simulation.

Reviewer #2: The authors have answered all the reviewer questions and handled the concerns.

**Have all data underlying the figures and results presented in the manuscript been provided?**

Reviewer #1: Yes

Reviewer #2: None

PLOS authors have the option to publish the peer review history of their article (what does this mean?). If published, this will include your full peer review and any attached files.

Reviewer #1: No

Reviewer #2: No

---

## [Editor Report · Acceptance letter]

21 Apr 2020

PCOMPBIOL-D-19-01313R1 

A Three-dimensional Phase-field Model for Multiscale Modeling of Thrombus Biomechanics in Blood Vessels

Dear Dr Karniadakis,

I am pleased to inform you that your manuscript has been formally accepted for publication in PLOS Computational Biology. Your manuscript is now with our production department and you will be notified of the publication date in due course.

With kind regards,

Laura Mallard
